# Deep Streaming View Clustering

Honglin Yuan [1]  Xingfeng Li [2]  Jian Dai [3]  Xiaojian You [1]  Yuan Sun [* 4 5]  Zhenwen Ren [* 2]

## Abstract

Existing deep multi-view clustering methods have demonstrated excellent performance, which addressing issues such as missing views and view noise. However, almost all existing methods are within a static framework, which assumes that all views have already been collected. Nevertheless, in practical scenarios, new views are continuously collected over time, which forms the stream of views. Additionally, there exists the data imbalance of distribution between different view streams, *i.e.*, concept drift problem. To this end, we propose a novel Deep Streaming View Clustering (DSVC) method, which mitigates the impact of concept drift on streaming view clustering (SVC). Specifically, DSVC consists of a knowledge base and three core modules. Through the knowledge aggregation learning module, DSVC extracts representative features and prototype knowledge from the new view. Subsequently, the distribution consistency learning module aligns the prototype knowledge from the current view with the historical knowledge distribution to mitigate the impact of concept drift. Then, the knowledge guidance learning module leverages the prototype knowledge to guide the data distribution and enhance the clustering structure. Finally, the prototype knowledge from the current view is updated in the knowledge base to guide the learning of subsequent views. Extensive experiments demonstrate that DSVC significantly outperforms state-of-the-art methods.

*Corresponding author [1]School of National Defense Science and Technology, Southwest University of Science and Technology, Mianyang, China [2]School of Computer Science and Technology,Southwest University of Science and Technology, Mianyang, China [3]Southwest Automation Research Institute, Mianyang, China [4]College of Computer Science, Sichuan University, Chengdu, China [5]National Key Laboratory of Fundamental Algorithms and Models for Engineering Numerical Simulation, Sichuan University, Chengdu, China. Correspondence to: Yuan Sun <sunyuan_work@163.com>, Zhenwen Ren <rzw@njust.edu.cn>.

*Proceedings of the 42nd International Conference on Machine Learning*, Vancouver, Canada. PMLR 267, 2025. Copyright 2025 by the author(s).

## 1. Introduction

With the advancement of information technology, data can often be obtained from multiple sources and views, which results in multi-view data. As a popular direction of unsupervised learning, multi-view clustering (MVC) aims to group data into distinct clusters by integrating information from different views, thereby uncovering shared clustering properties of multi-view data (Qin et al., 2024; Sun et al., 2024b; Xu et al., 2022; Song et al., 2025; Sun et al., 2024d; Wong et al., 2023; Sun et al., 2024a). In contrast to single-view clustering, MVC can capture data information from multiple perspectives, which provides a more comprehensive understanding.

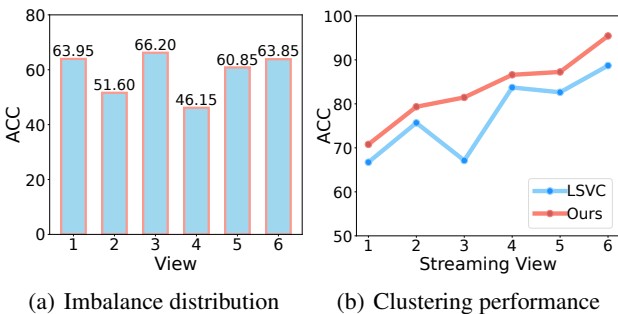

(a) Imbalance distribution    (b) Clustering performance

*Figure 1.* The illustration of the concept drift problem on the Handwritten dataset. (a) is the clustering performance of each view. Due to distribution imbalance across different views, which leads to different views exhibiting varying performance. This indicates the presence of concept drift across different views. (b) is the experiment of view streaming increases. Due to concept drift problem among different views, the clustering performance of LSVC exhibits continuous fluctuations as more views are incrementally collected. It shows that existing lifelong learning methods (LSVC) are unable to address the concept drift problem in SVC tasks.

Over the last decade, researchers have proposed a significant number of MVC methods. Traditional MVC methods can generally be classified into four main types, *i.e.*, non-negative matrix factorization (NMF) (Feng et al., 2024; Li et al., 2023b), graph-based learning (Du et al., 2023; Liang et al., 2024; Liu et al., 2023; Li et al., 2025), multi-kernel-based (Xu et al., 2024b; Li et al., 2022), and subspace-based methods (Li et al., 2024a; Sun et al., 2021). Thanks to the powerful nonlinear representation capabilities of deep learn-

ing, a large number of deep multi-view clustering (DMVC) (Bian et al., 2024; Chen et al., 2024; Yuan et al., 2024; Fu et al., 2025a; Yuan et al., 2025) methods have been proposed in recent years, which have gradually emerged as a mainstream research direction. Although existing MVC methods have demonstrated excellent clustering performance within static frameworks, they often overlook the treatment of real-time data. In dynamic task environments, the new view will continuously accumulate over time, *i.e.*, streaming view data. In this context, the model should be updated and trained on new data promptly, rather than waiting for the complete collection of all view data before training. This limitation poses a significant challenge to the processing of streaming view data.

To tackle this challenge, LSVC (Li et al., 2024b) employs the lifelong learning paradigm, which utilizes a consensus bipartite graph to align the current view knowledge with the historical knowledge, and fuse the current view knowledge into the historical knowledge, thereby enabling streaming view clustering (SVC). However, since data from different views originate from distinct sensors or angles, there exists an imbalance situation in the distribution of the data across view streams, such situation is known as concept drift (Sharief et al., 2024; Jiao et al., 2022), as shown in Fig.1(a). Due to the issue of concept drift, the outdated model struggles to adapt to the training of new view data, thereby leading to a significant decline in clustering performance. Specifically, as illustrated in Fig.1(b), LSVC places excessive emphasis on preserving historical knowledge, in which outdated information can mislead model training, making it difficult to adapt to and effectively process new view data. Therefore, it is essential to overcome concept drift and ensure the collected data exhibits a unified feature representation and distribution.

In this paper, we propose a novel deep streaming view clustering (DSVC) method, as illustrated in Fig.2. DSVC consists of a knowledge base and three core modules: the knowledge aggregation learning module (KAL), the distribution consistency learning module (DCL), and the knowledge guidance learning module (KGL). Specifically, when a new view arrives, we utilize the KAL module to aggregate representative prototype knowledge and feature representations from the current view. Then, the DCL module employs distribution consistency learning loss to align prototype knowledge of the current view with historical knowledge, which promotes distribution consistency of collected data. Subsequently, the KGL module leverages prototype knowledge to enhance the clustering structure of the features. In general, the main contributions of our work are as follows:

- We propose a novel deep streaming multi-view clustering framework named Deep Streaming View Clustering (DSVC). To the best of our knowledge, our DSVC

could be the first work to reveal and study the issue of concept drift in the context of streaming view clustering.

- To mitigate the effect of concept drift, we propose a distributional consistency learning module that aligns the prototype knowledge of the current view with historical knowledge distribution, with the aim of improving the consistency of the collected data distribution.

- We propose a knowledge guidance learning module, which leverages prototype knowledge to guide data distribution and enhance the clustering structure of the feature representation.

- Experimental results on eight datasets demonstrate that our proposed DSVC significantly outperforms 13 state-of-the-art MVC methods, highlighting its effectiveness in real-world streaming view scenarios.

## 2. Related Work

### 2.1. Deep Multi-view Clustering

Due to the powerful nonlinear representation capabilities of deep neural networks, many deep multi-view clustering (DMVC) algorithms (Fu et al., 2025b; Cui et al., 2024; Sun et al., 2024c) have been proposed in recent years. Existing DMVC methods primarily address three key challenges: the partial view misalignment problem, the partial view missing problem, and the noisy view problem. Specifically, to address the issue of misaligned views in multi-view data, DealMVC (Yang et al., 2023) introduces a dual-contrast alignment network, which promotes consistency between paired view features by integrating both global and local alignment losses. SURE (Yang et al., 2022) introduces a noise-robust contrastive loss, which effectively mitigates the effect of false negatives due to random sampling. To address the problem of missing partial views, ProImp (Li et al., 2023a) and CPSPAN (Jin et al., 2023) leverage the relationship between prototypes and samples to infer missing data. To address the challenge of noisy data in the views, MVCAN (Xu et al., 2024a) mitigates the side effects of noisy views, which by supporting the non-shared parameters for multiple views and inconsistent clustering predictions. RMCNC (Sun et al., 2024c) introduces a noise-tolerance multi-view contrastive loss that avoids overemphasizing noisy data, thereby alleviating issues associated with data noise. However, in practical scenarios, multi-view data is often collected in the form of streaming views. Existing deep learning methods fail to account for this dynamic setting. To address this limitation, we propose a deep streaming view clustering (DSVC) method tailored for dynamic clustering tasks in real-world applications.

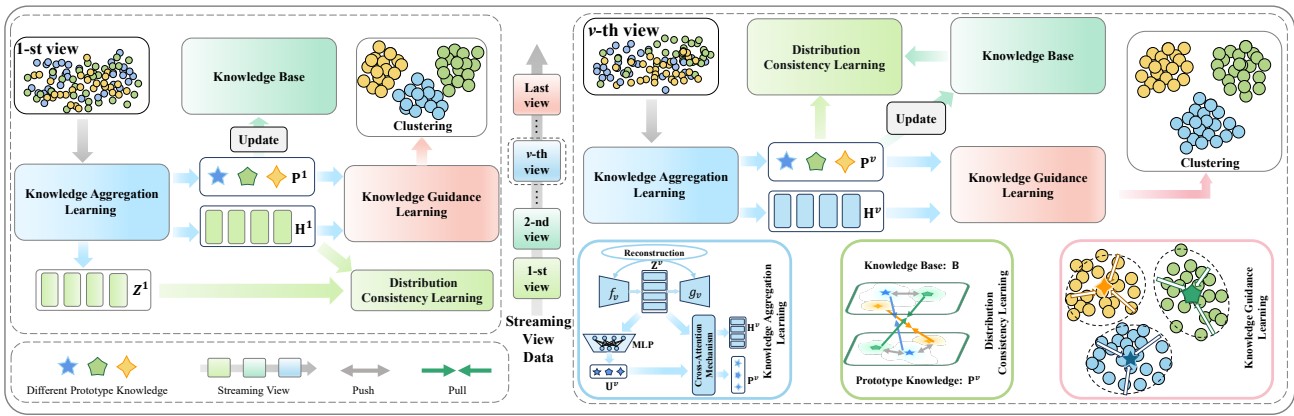

*Figure 2.* The framework of the proposed DSVC. We employ the knowledge aggregation learning module to extract prototype knowledge and feature representations from the current view. The distribution consistency learning module is utilized to align the knowledge distribution of the current view with that of the historical knowledge, thereby mitigating the influence of concept drift. Furthermore, the knowledge guidance learning module is introduced to utilize prototype knowledge to guide data distribution and strengthen the clustering structure. Finally, the prototype knowledge from the current view is updated into the knowledge base, which guides the training of subsequent views.

## 2.2. Stream Learning

Streaming learning (also called continual learning or incremental learning) is a machine learning paradigm, which aims to process data points or data blocks that arrive sequentially over time. Existing streaming learning scenarios include task-stream learning, domain-stream learning, and class-stream learning. Specifically, task-stream learning (Oren & Wolf, 2021; Jiang & Celiktutan, 2023; Li et al., 2024c) aims to process new tasks that arrive sequentially, where the new tasks significantly differ from the historical ones. In domain-stream learning (Wang et al., 2024; Shi & Wang, 2023; Pei et al., 2024), the sample categories across different domains remain the same, but the input distributions are different. In class-stream learning (Zhou et al., 2024b;a; Huang et al., 2024), the model is required to continuously distinguish between newly arrived classes and previously learned classes. Overall, existing streaming learning methods overlook the following issues: (1) With the development of multi-view learning, data is often continuously accumulated in the form of view streams. However, existing methods rarely take this scenario into account. (2) Due to the fact that data from different views originate from different sensors or views, there is a situation of imbalance and instability in the distribution of data across view streams (*i.e.*, concept drift), which presents a new challenge for the adaptability ability of the model to deal with the new view data. For those, we propose a novel DSVC method to enable streaming view clustering and mitigate the effect of concept drift on model performance.

## 3. Method

### 3.1. Notation

We assume that the dataset $\{\mathbf{X}^i\}_{i=1}^V \in \mathbb{R}^{N \times D^v}$ consists of $V$ views, with the dimensionality of $D^v$ and $N$ samples. During training, the $V$ views arrive sequentially, and data from views where $i < v$ is not accessible to the model during the training of the $v$-th view $\mathbf{X}^v = \{x_1^v, x_2^v, \ldots, x_N^v\} \in \mathbb{R}^{N \times D^v}$. $\mathbf{Z}^v \in \mathbb{R}^{N \times d}$ denotes the feature representation, where $d$ is the feature dimensionality. $\mathbf{H}^v \in \mathbb{R}^{N \times d}$ represents the reconstructed feature representation. $\mathbf{P}^v \in \mathbb{R}^{K \times d}$ refers to the prototype knowledge of view $v$ in $\mathbf{X}^v$, and $K$ indicates the number of prototype knowledge.

### 3.2. The Objective Function

The overall architecture of our proposed method is illustrated in Fig.2. When collecting the first view data, the knowledge aggregation learning (KAL) module preserves as much view information as possible through reconstruction loss. It then leverages an attention mechanism to reconstruct features and prototype knowledge. However, at this stage, the knowledge base is empty and cannot be used to constrain the knowledge distribution of the current view. To ensure that the reconstructed features remain aligned with the original data distribution, we employ the distribution consistency learning (DCL) module. This module promotes distribution consistency between the latent features and the reconstructed features, thereby enhancing the representativeness of the prototype knowledge learned from the first view. In addition, we leverage the knowledge guidance learning (KGL) module to guide the data distribution, thereby strengthening the clustering structure. Finally, the

prototype knowledge from the first view is updated into the knowledge base. Consequently, the total loss for the first view is formulated as:

$$\mathcal{L} = \alpha\mathcal{L}_r + \mathcal{L}_d + \beta\mathcal{L}_g. \tag{1}$$

When the number of collected views exceeds one, we continue to utilize the KAL module to extract feature representations and prototype knowledge. However, because of concept drift exists between view streams, which makes it difficult for previous models to adapt to the training requirements of subsequent views, as shown in Fig.1(b). To address this issue, the distribution consistency learning module aligns the prototype knowledge of the current view with the knowledge base, which mitigates the effect of concept drift. Additionally, we leverage the KGL module to enhance the clustering structure. Finally, the prototype knowledge of the current view is updated into the knowledge base to guide the training of subsequent views. Overall, the total loss is defined as:

$$\mathcal{L} = \alpha\mathcal{L}_r + \mathcal{L}_d + \beta\mathcal{L}_g, \tag{2}$$

where $\mathcal{L}_r$ represents the reconstruction loss, $\mathcal{L}_d$ represents the distribution consistency learning loss, and $\mathcal{L}_g$ represents the knowledge guidance learning loss. $\alpha$ and $\beta$ are trade-off parameters.

### 3.3. Knowledge Aggregation Learning

The collected data often contains substantial redundant information. To obtain cluster-friendly feature representations, we employ the autoencoder to extract the latent feature representations of the data. Specifically, we employ the encoder $f_{\theta^v}^v$ to extract features $\mathbf{Z}^v \in \mathbb{R}^{N \times d}$ from the data $\mathbf{X}^v \in \mathbb{R}^{N \times D^v}$. Then, the decoder $g_{\phi^v}^v$ reconstructs the data $\tilde{\mathbf{X}}^v \in \mathbb{R}^{N \times D^v}$ from the features $\mathbf{Z}^v$. Mathematically, the reconstruction loss is defined as:

$$\mathcal{L}_r = \left\| \mathbf{X}^v - g_{\phi^v}^v(f_{\theta^v}^v(\mathbf{X}^v)) \right\|_2^2, \tag{3}$$

where $\theta^v$ and $\phi^v$ represent the network parameters of the encoder and decoder, respectively. To capture the commonalities among the data and the unique characteristics between classes, we first utilize MLP to derive $K$ prototype knowledge $\mathbf{U}^v = W(\mathbf{Z}^v) \in \mathbb{R}^{N \times K}$ from latent features $\mathbf{Z}$, where $W$ represents the linear layer of the MLP network, $K = M * C$ with $M \in \mathbb{N}_+$ is a hyper-parameter and $C$ indicates the number of classes. However, the capability of the MLP depends on its depth and width, which makes it challenging to extract high-quality shared knowledge from complex latent representations. Therefore, we employ the cross-attention mechanism to adjust the distribution among the data using prototype knowledge, and aggregate more discriminative prototype knowledge from the features. Specifi-

cally, the cross-attention $\mathbf{A}^v$ is calculated as follows:

$$\mathbf{A}^v = \text{Softmax}\left((W_Z^v(\mathbf{Z}^v)^T W_U^v(\mathbf{U}^v)/\sqrt{d}\right), \tag{4}$$

where $W_Z^v$ and $W_U^v$ represent the linear layers for the latent features and prototype knowledge in the $v$-th view, respectively. Subsequently, we use the attention $\mathbf{A}^v$ to re-aggregate prototype knowledge $\mathbf{P}^v \in \mathbb{R}^{N \times K}$ from the latent features, and reconstruct the features $\mathbf{H}^v \in \mathbb{R}^{N \times d}$ from the prototype knowledge. Mathematically, it can be expressed as follows:

$$\begin{aligned} \mathbf{P}^v &= \mathbf{U}^v + \mathbf{A}^v W'^v_H \mathbf{Z}^v, \\ \mathbf{H}^v &= \mathbf{Z}^v + \mathbf{A}^v W'^v_P \mathbf{U}^v, \end{aligned} \tag{5}$$

where $W'_P$ and $W'_Z$ represent additional linear layers for the prototype knowledge and latent features, respectively.

### 3.4. Distribution Consistency Learning

During the data collection process, there exists the concept drift between the stream views. To this end, we propose a distribution consistency learning strategy that aligns the knowledge of the current view with historical knowledge $\mathbf{B} = \{b_1, b_2, \ldots, b_K\} \in \mathbb{R}^{K \times d}$, thereby promoting the consistency of the collected data distribution. Specifically, we use cosine similarity to evaluate the correlation between current view knowledge and historical knowledge. The formula is defined as follows:

$$S(p_i^v, b_j) = \frac{p_i^v(b_j)^T}{\|p_i^v\|\|b_j\|}, \tag{6}$$

where $p_i^v$ represents the $i$-th prototype knowledge of the current view, and $b_j$ denotes the $j$-th knowledge of the historical view. Thus, the probability that the two types of knowledge belong to the same class in the historical view is given by:

$$Q_i^p = \frac{\exp\left(S(b_i, p_i^v)/\tau\right)}{\sum_{j=1}^K \exp\left(S(b_i, p_i^v)/\tau\right) + \sum_{j \neq i}^K \exp\left(S(p_i, p_j)/\tau\right)}, \tag{7}$$

where $\tau$ represents the temperature parameter, which is set to 1. Additionally, we normalize $Q^b$ to ensure that $\sum_{i=1}^K Q_i^b = 1$. To mitigate concept drift and promote consistency in sample distributions in different view streams, we propose the distribution consistency learning loss, which is defined as follows:

$$\begin{aligned} \mathcal{L}_d &= \frac{1}{2}[D_{KL}(Q^b \| Q^p) + D_{KL}(Q^p \| Q^b)] \\ &= \frac{1}{2}\sum_{i=1}^K [Q_i^b \log \frac{Q_i^b}{Q_i^p} + Q_i^p \log \frac{Q_i^p}{Q_i^b}], \end{aligned} \tag{8}$$

where $D_{KL}$ represents the Kullback-Leibler divergence. The goal of $\mathcal{L}_d$ is to maximally align the knowledge distribution between the current view and the historical view,

thereby unifying the data distribution. Moreover, when the first view is collected, the knowledge base is empty, therefore it cannot be used to constrain the knowledge distribution of the current view. To ensure that the reconstructed features do not deviate from the original data distribution, we utilize the loss defined in Eq.(8), which aims to promote consistency between the latent and reconstructed features. This helps to make the prototype knowledge learned from the first view more representative.

### 3.5. Knowledge Guidance Learning

To optimize the feature distribution and strengthen the clustering structure, we use the aligned prototype knowledge to guide the data distribution of the current view. Specifically, we first compute the similarity between the features and the prototype knowledge. The formula is as follows:

$$\mathbf{M}(h_i^v, p_j^v) = \exp(S\langle h_i^v, p_j^v\rangle), \qquad (9)$$

where $h_i^v$ represents the $i$-th feature of the current view, and $p_j^v$ denotes the $j$-th prototype knowledge. Thus, the probability that the $j$-th prototype knowledge contains the $i$-th feature is given by:

$$G_{(i,j)} = \frac{\exp(S\langle h_i^v, p_j^v\rangle)}{\sum_{k=1}^{K} \exp(S\langle h_i^v, p_k^v\rangle)}, \qquad (10)$$

where $K$ is the number of prototype knowledge. To optimize the clustering structure of the features, our knowledge guidance learning loss is defined as follows:

$$\mathcal{L}_g = \frac{1}{KN} \sum_{j=1}^{K} \sum_{i=1}^{N} \log G_{(i,j)}, \qquad (11)$$

where $N$ is the number of features. Finally, we update the prototype knowledge of the current view into the knowledge base ( $i.e.$, $\mathbf{B} = \mathbf{P}^v$ ), to guide the training of the subsequent views collected.

## 4. Experiments

### 4.1. Datasets

We design a series of experiments on eight widely used datasets, which encompass various data types, to demonstrate the effectiveness of our DSVC method. Detailed information for all datasets is provided in Tab.1. Concretely, **ALOI-10** (Geusebroek et al., 2005) contains 1,079 samples across 10 categories, with each sample having four features, $i.e.$, HSB, RGB, Colorsim, and Haralick. **HandWritten** (LeCun et al., 1989) consists of 2,100 samples, covering 10 categories corresponding to digits 1 - 9. Each sample is characterized by six types of features including Pixel, Fourier, Profile, Zer, Kar, and Mor. **LandUse-21** (Yang & Newsam, 2010) contains 2,100 satellite images across 21 categories. It

includes three views, $i.e.$, GIST, PHOG, and LBP. **Scene-15** (Fei-Fei & Perona, 2005) dataset consists of 15 categories, including scenes such as office, kitchen, living room, and mountain, with a total of 4,485 samples. Each sample is represented by three distinct features, $i.e.$, GIST, PHOG, and LBP. **ALOI-100** (Geusebroek et al., 2005) consists of 10,800 object images, belonging to 100 classes. Multi-view data is constructed by extracting HSB, RGB, Colorsim, and Haralick features from these images, respectively. **Stl10-fea** (Coates et al., 2011) comprises 13,000 samples spanning 10 categories. Our experiments use its three distinct deep features as the different views. **YoutubeFace** (Wolf et al., 2011) consists of 101,499 samples across 31 classes. We utilize its Audio volume, Cuboids histogram, Vision HIST, HOG, and MISC features for experimental evaluation. **ALOI-1000** (Geusebroek et al., 2005) comprises 110,250 object images, and 1,000 classes. Each sample has four features, $i.e.$, HSB, RGB, Colorsim, and Haralick.

Table 1. Multi-view datasets in the experiment.

| Datasets | Samples | Clusters | Views | Dimensionality |
|---|---|---|---|---|
| ALOI-10 | 1079 | 10 | 4 | 64/64/77/13 |
| HandWritten | 2000 | 10 | 6 | 216/76/64/6/240/47 |
| LandUse-21 | 2100 | 21 | 3 | 20/59/40 |
| Scene-15 | 4485 | 15 | 3 | 20/59/40 |
| ALOI-100 | 10800 | 100 | 4 | 77/13/64/125 |
| Stl10-fea | 13000 | 10 | 3 | 1024/512/2048 |
| YoutubeFace | 101499 | 31 | 5 | 64/512/64/647/838 |
| ALOI-1000 | 110250 | 1000 | 4 | 125/77/13/64 |

### 4.2. Comparison Methods

We compare DSVC with 12 state-of-the-art DMVC methods and the sole SVC method to demonstrate its effectiveness. Specifically, these methods include: COMIC (Peng et al., 2019), DSRL (Wang et al., 2021), MFLVC (Xu et al., 2022), DCP (Lin et al., 2022), CVCL (Chen et al., 2023), DealMVC (Yang et al., 2023), CPSPAN (Jin et al., 2023), DMCE (Zhao et al., 2023), GDMVC (Bai et al., 2024), MAGA (Bian et al., 2024), FMCSC (Chen et al., 2024), MVCAN (Xu et al., 2024a), and LSVC (Li et al., 2024b).

To comprehensively evaluate these comparison methods, we employ three widely used metrics: Accuracy (ACC), Normalized Mutual Information (NMI), and Adjusted Rand Index (ARI).

### 4.3. Experimental Settings

Our DSVC is implemented in PyTorch 2.3.0, and all experiments are performed on a Linux system with an NVIDIA GPU and 32GB RAM. For our DSVC, the autoencoder consists of a fully connected network. The encoder ( decoder) network has the architecture of $D^v - 512 - 1024 - 512 - 256$ ($256 - 512 - 1024 - 512 - D^v$), where $D^v$ represents the feature dimension of each view stream. In the experiments,

*Table 2.* Performance comparison (mean ± standard deviation) with four datasets, 'O/M' represents out-of-memory, '*' represents the streaming view clustering method, the optimal results are highlighted in bold **red** and the suboptimal results are shown in bold **blue**.

| Datasets | HandWritten | | | ALOI-10 | | | Landuse-21 | | | Scene-15 | | |
|---|---|---|---|---|---|---|---|---|---|---|---|---|
| Evaluation metrics | ACC | NMI | ARI | ACC | NMI | ARI | ACC | NMI | ARI | ACC | NMI | ARI |
| COMIC(ICML'19) | 63.60±0.43 | 62.91±0.66 | 63.91±0.36 | 56.23±2.62 | 60.43±2.40 | 66.58±2.65 | 21.91±1.25 | 28.58±0.92 | 8.93±0.54 | 33.89±0.64 | 40.55±0.48 | **24.16±0.91** |
| DSRL(TPAMI'21) | 85.78±0.29 | 88.22±0.48 | 81.17±0.64 | 78.76±2.60 | 65.83±1.84 | 61.28±4.10 | 28.23±0.75 | 28.79±0.61 | 13.06±0.04 | 45.53±0.86 | 40.11±0.54 | 24.01±0.67 |
| MFLVC(CVPR'22) | 83.74±4.09 | 80.83±2.49 | 81.95±4.02 | 63.66±1.01 | 67.23±2.61 | 60.27±1.65 | 19.90±0.28 | 22.33±0.40 | 7.13±0.21 | 37.73±1.04 | 37.81±0.91 | 21.56±1.05 |
| DCP(TPAMI'22) | 78.14±0.45 | 76.86±0.24 | 68.65±0.52 | 64.50±4.66 | 63.60±3.51 | 55.99±4.09 | **28.29±1.36** | 32.49±1.20 | **14.01±0.67** | **42.27±3.23** | **41.05±2.66** | 23.23±2.06 |
| CVCL(ICCV'23) | 84.55±9.43 | 87.29±3.51 | 80.96±7.75 | 82.33±8.47 | 80.34±6.51 | 71.94±10.75 | 19.20±0.62 | 21.95±0.23 | 6.43±0.18 | 36.15±2.73 | 38.19±1.83 | 20.69±1.84 |
| DealMVC(MM'23) | 80.69±0.48 | 80.13±0.46 | 71.79±0.61 | 71.38±4.49 | 75.40±1.98 | 59.86±5.05 | 19.01±0.80 | 16.75±1.24 | 13.50±0.49 | 35.79±0.94 | 36.59±0.81 | 21.97±0.83 |
| CPSPAN(CVPR'23) | 85.72±4.68 | 84.52±2.35 | 75.77±3.62 | 67.84±7.71 | 81.01±2.96 | 63.91±7.64 | 25.59±1.55 | 32.49±1.35 | 12.00±0.67 | 38.78±2.44 | 37.71±2.23 | 21.59±2.17 |
| DMCE(23'PR) | **94.30±3.97** | **91.35±1.77** | **90.06±3.91** | 61.33±2.06 | 71.04±1.01 | 52.73±1.71 | 20.31±0.70 | 28.19±0.46 | 8.47±0.83 | 32.37±2.48 | 31.67±1.16 | 15.62±1.24 |
| GDMVC(KBS'24) | 84.32±0.34 | 88.43±0.45 | 81.78±0.63 | **85.64±4.08** | 83.78±1.87 | 71.87±4.83 | 27.21±1.38 | **33.46±1.61** | 12.34±0.45 | 38.55±1.16 | 39.21±0.38 | 20.33±0.49 |
| MAGA(IF'24) | 92.22±1.04 | 84.95±1.42 | 83.63±2.07 | 57.44±8.40 | 62.64±6.61 | 43.57±8.50 | 19.46±1.01 | 20.18±0.75 | 6.31±0.55 | 32.21±0.78 | 34.50±0.75 | 17.68±0.58 |
| FMCSC(NeurIPS'24) | 84.35±0.38 | 74.01±0.22 | 70.41±0.29 | 85.17±3.82 | **83.98±1.05** | **77.53±2.29** | 21.66±0.36 | 23.25±0.53 | 8.20±0.26 | 34.29±2.67 | 31.57±2.08 | 18.23±1.95 |
| MVCAN(CVPR'24) | 94.12±0.57 | 88.41±0.59 | 87.51±1.12 | 54.51±3.81 | 63.27±2.51 | 45.08±3.55 | 22.38±1.21 | 28.94±1.50 | 9.45±0.688 | 37.76±1.06 | 39.18±0.93 | 20.94±0.92 |
| LSVC(TNNLS'24) * | 88.69±0.87 | 89.45±0.79 | 81.47±0.77 | 66.90±4.01 | 67.81±3.43 | 60.53±4.12 | 24.66±1.44 | 26..71±0.58 | 11.13±0.67 | 40.68±1.26 | 37.54±1.31 | 23.98±1.66 |
| **DSVC(Ours) *** | **95.44±0.32** | **90.26±0.61** | **90.16±0.68** | **86.69±3.85** | **85.67±2.94** | **80.37±3.46** | **28.52±1.16** | **33.09±0.74** | **14.16±0.83** | **45.33±0.88** | **43.13±1.13** | **27.61±1.10** |

*Table 3.* Performance comparison (mean ± standard deviation) with four datasets, 'O/M' represents out-of-memory, '*' represents the streaming view clustering method, the optimal results are highlighted in bold **red** and the suboptimal results are shown in bold **blue**.

| Datasets | Stl10-fea | | | ALOI-100 | | | YoutubeFace | | | ALOI-1000 | | |
|---|---|---|---|---|---|---|---|---|---|---|---|---|
| Evaluation metrics | ACC | NMI | ARI | ACC | NMI | ARI | ACC | NMI | ARI | ACC | NMI | ARI |
| COMIC(ICML'19) | 20.51±3.52 | 12.66±3.07 | 15.33±2.40 | 59.19±1.93 | 59.83±1.44 | 54.69±0.78 | O/M | O/M | O/M | O/M | O/M | O/M |
| DSRL(TPAMI'21) | O/M | O/M | O/M | O/M | O/M | O/M | O/M | O/M | O/M | O/M | O/M | O/M |
| MFLVC(CVPR'22) | 35.92±8.47 | 19.91±9.28 | 14.89±8.84 | 66.23±1.50 | 67.14±1.33 | 56.77±1.09 | **24.54±1.11** | 21.62±1.04 | **4.56±0.58** | O/M | O/M | O/M |
| DCP(TPAMI'22) | 20.41±1.97 | 14.44±2.81 | 16.90±1.78 | 60.42±0.52 | 81.98±0.61 | 51.12±0.49 | 21.92±1.61 | 18.52±1.23 | 3.84±0.54 | **48.12±0.86** | 66.01±0.72 | **37.50±0.65** |
| CVCL(ICCV'23) | 20.68±0.54 | 14.84±0.24 | 22.82±1.56 | 60.26±2.54 | 82.63±0.67 | 54.67±1.50 | 19.74±3.27 | 19.82±0.91 | 4.22±0.45 | O/M | O/M | O/M |
| DealMVC(MM'23) | 12.14±0.33 | 14.24±0.91 | 14.41±0.44 | 19.66±1.52 | 58.92±0.51 | 12.16±0.83 | 18.95±0.75 | 15.64±0.74 | 3.32±0.50 | O/M | O/M | O/M |
| CPSPAN(CVPR'23) | 24.46±10.59 | 15.47±12.59 | 10.02±9.24 | 66.77±2.05 | 83.63±0.95 | 56.59±2.99 | 20.63±3.24 | 18.76±3.19 | 4.02±1.19 | 43.81±2.61 | **76.12±1.37** | 30.86±2.82 |
| DMCE(23'PR) | 28.14±4.02 | 19.59±3.54 | 11.14±3.03 | 74.83±0.52 | 83.82±0.27 | 62.20±0.50 | O/M | O/M | O/M | O/M | O/M | O/M |
| GDMVC(KBS'24) | 14.33±0.69 | 14.43±0.65 | 6.8±1.32 | **80.15±0.66** | **86.82±0.26** | **64.13±0.85** | O/M | O/M | O/M | O/M | O/M | O/M |
| MAGA(IF'24) | **56.37±16.12** | **52.54±15.56** | **42.03±14.55** | 52.98±0.91 | 69.57±0.57 | 40.45±48 | 22.95±1.36 | **23.68±1.21** | 3.77±0.37 | O/M | O/M | O/M |
| FMCSC(NeurIPS'24) | 25.04±2.63 | 17.52±3.26 | 28.95±2.36 | 57.06±1.70 | 65.90±0.64 | 57.46±0.12 | 22.46±1.01 | 21.10±0.89 | 4.14±0.25 | O/M | O/M | O/M |
| MVCAN(CVPR'24) | 48.57±12.44 | 52.24±16.34 | 39.41±13.07 | 66.23±0.94 | 82.96±0.42 | 55.66±0.42 | 12.76±0.36 | 13.65±0.49 | 2.071±0.14 | O/M | O/M | O/M |
| LSVC(TNNLS'24) * | 47.98±6.31 | 51.03±6.47 | 40.90±4.16 | 64.11±0.66 | 74.15±0.48 | 55.01±1.49 | O/M | O/M | O/M | O/M | O/M | O/M |
| **DSVC(Ours) *** | **63.41±9.96** | **62.66±9.04** | **53.58±9.72** | **80.53±0.59** | **89.01±0.18** | **73.96±0.82** | **25.09±1.09** | **24.45±0.85** | **5.60±0.41** | **58.32±0.56** | **81.20±0.24** | **44.93±0.49** |

*Table 4.* Ablation studies on eight datasets, where '√' indicates the used component.

| Datasets | | | ALOI-10 | | HandWritten | | Landuse-21 | | Scene-15 | | Stl10-fea | | ALOI-100 | | YoutubeFace | | ALOI-1000 | |
|---|---|---|---|---|---|---|---|---|---|---|---|---|---|---|---|---|---|---|
| $\mathcal{L}_r$ | $\mathcal{L}_d$ | $\mathcal{L}_g$ | ACC | NMI | ACC | NMI | ACC | NMI | ACC | NMI | ACC | NMI | ACC | NMI | ACC | NMI | ACC | NMI |
| √ | | | 69.05 | 74.90 | 81.75 | 81.94 | 18.00 | 28.73 | 30.77 | 33.30 | 38.32 | 34.48 | 58.70 | 81.27 | 18.37 | 19.60 | 36.71 | 73.57 |
| √ | √ | | 74.42 | 80.84 | 86.45 | 78.29 | 24.86 | 32.23 | 39.60 | 40.68 | 52.65 | 58.36 | 79.05 | 88.5 | 23.50 | 23.04 | 54.13 | 76.35 |
| √ | | √ | 76.92 | 80.08 | 88.33 | 86.70 | 24.75 | 29.86 | 40.02 | 41.50 | 53.75 | 52.66 | 77.33 | 87.16 | 23.07 | 23.04 | 56.00 | 79.95 |
| | √ | √ | 78.96 | 80.25 | 90.95 | 83.44 | 24.62 | 31.67 | 35.72 | 36.93 | 45.28 | 38.53 | 54.43 | 71.16 | 18.45 | 19.81 | 40.42 | 71.40 |
| √ | √ | √ | **86.69** | **85.67** | **95.44** | **90.26** | **28.52** | **33.09** | **45.33** | **43.13** | **63.41** | **62.66** | **80.53** | **89.01** | **25.09** | **24.24** | **58.32** | **81.20** |

we train each collected view for 200 epochs with batch size 256 and learning rate 0.0001. Additionally, we use Adam optimizer for model optimization and employ ReLU as the activation function.

In DSVC, which included two adjustable parameters, *i.e.*, $\alpha$ and $\beta$. For the HandWritten and Scene-15 datasets, we set $\alpha$ and $\beta$ to 1 and 0.1, respectively. For the Stl10-fea dataset, $\alpha$ and $\beta$ are set to 0.001 and 1. For all other datasets, we uniformly set $\alpha$ and $\beta$ to 0.1. To comprehensively evaluate our clustering performance, we tested all methods using five different random seeds and calculated the mean and standard deviation as the final results.

## 4.4. Experimental Results Analysis

Tabs.2 and 3 present the clustering results of DSVC compared to 12 state-of-the-art DMVC methods and the sole

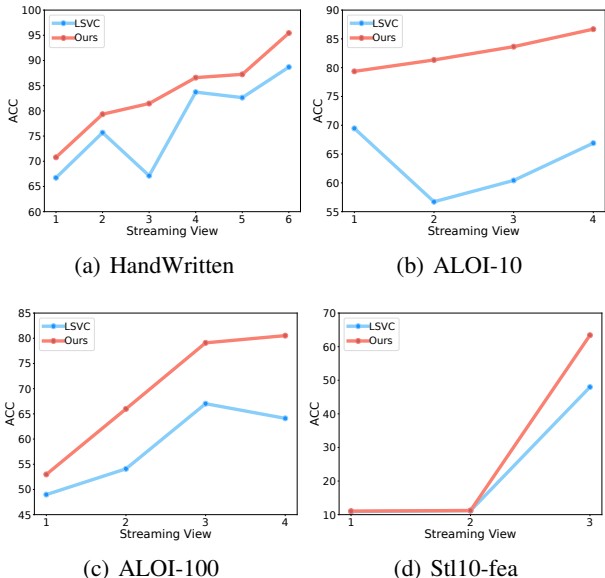

*Figure 3.* The changes of clustering performance on four datasets, when view data continue to accumulate.

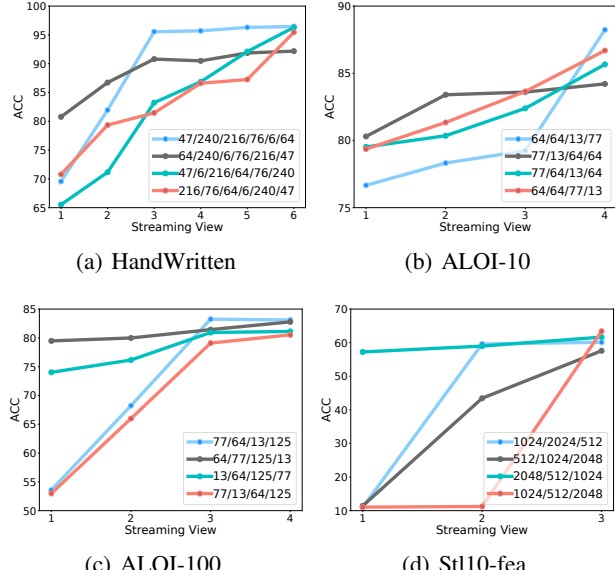

*Figure 4.* The clustering performance with different sequences of views streaming on four datasets

SVC method across three evaluation metrics on eight different datasets. From the tables, the following observations can be obtained:

- Obviously, our DSVC demonstrates superior performance compared to the other 12 state-of-the-art DMVC methods and the sole SVC method in nearly all scenarios. Specifically, we take the ACC metric as an example, on the eight datasets (from ALOI-10 to ALOI-1000), DSVC outperforms the suboptimal method by improvements of 1.14%, 1.05%, 0.23%, 3.06%, 7.04%, 0.38%, 0.55%, and 10.20%, respectively. This demonstrates that DSVC can extract representative information from the continuously evolving view streams, thereby uncovering more optimal clustering attributes.

- In addition to achieving superior clustering performance, DSVC exhibits greater stability than suboptimal methods, as evidenced by the lower standard deviation. This is because our knowledge aggregation learning (KAL) module effectively extracts representative prototype knowledge and feature representations from the current view data. The distribution consistency learning (DCL) module aligns the prototype knowledge of the current view with the historical knowledge distribution, thereby mitigating the effect of concept drift. The knowledge guidance learning (KGL) module leverages aligned prototype knowledge to constrain the data distribution of the current view, which enhances the clustering structure. The three modules work synergistically to achieve the most stable performance.

- As the scale of datasets continues to grow, LSVC and some DMVC methods are unable to handle these large-scale datasets within the constraints of limited computational resources. For example, the ALOI-1000 dataset can only be processed by the two DMVC methods ( *i.e.*, CPSPAN and DCP ) and our DSVC. However, our approach is not only capable of handling these large-scale datasets, but also achieves optimal performance. This is because DSVC is specifically designed to tackle the challenge of streaming view clustering in real-world tasks. Therefore, only one view is considered at a time during each training step instead of training on all views simultaneously, which significantly reduces the computational overhead.

## 4.5. View Stream Analysis

To comprehensively analyze the performance of DSVC on dynamic tasks, we present the clustering results after processing new views, as well as its performance across different view streaming sequences. Specifically, as shown in Fig.3, with the continuous accumulation of new views, LSVC exhibits unstable performance, while the performance of our method is persistent increase. This is attributed to the problem of concept drift across different view streams, which results in the previous model being unsuitable for training the next view. In contrast, our DSVC aligns the prototype knowledge of the current view with the historical knowledge by DCL, which unifies the data distribution across different view streams and effectively mitigates the concept drift problem. Compared to DMVC

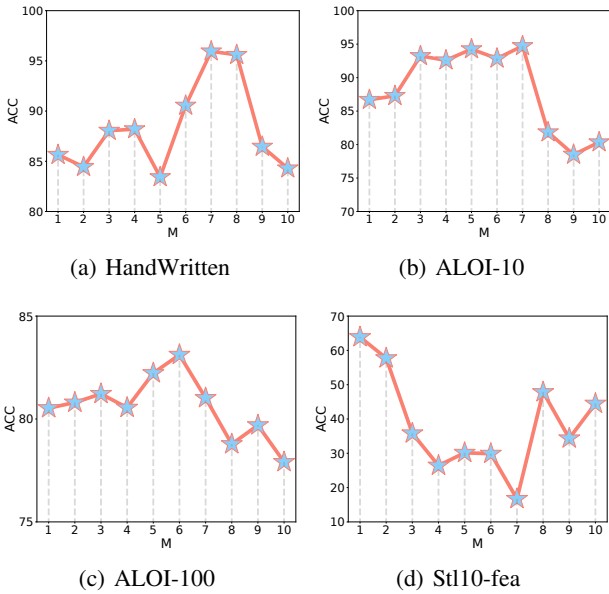

Figure 5. The clustering performance with different prototype knowledge numbers on four datasets

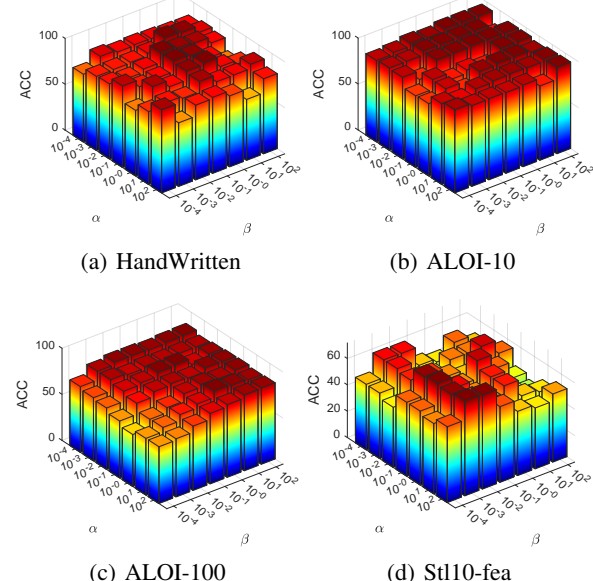

Figure 6. Parameter sensitivity analysis on eight datasets.

methods, our DSVC consistently achieves near-optimal performance. This is due to the KGL, which leverages prototype knowledge to guide the data distribution and enhance the clustering structure.

Additionally, as shown in Fig.4, the initial performance gap is quite significant when processing view streams of different sequences. However, as the views accumulate, the performance gap between different sequences gradually decreases. This is because the initially collected views exhibit significant differences in data distribution, which leads to a noticeable performance gap. As more views are accumulated, our DCL module progressively aligns the data distribution, while the KGL module continuously enhances the clustering structure. As a result, the final performance tends to be stable and consistent. For more experiments on view flow analysis, please refer to Section B of the appendix.

### 4.6. Prototype Knowledge Number Analysis

To evaluate the impact of the number of prototype knowledge on the clustering performance of the model, we conduct the experiment with prototype knowledge numbers ranging from $K$ to $M * K$, where $M = 1, 2, ..., 10$. As shown in Fig.5, the clustering performance can be enhanced with an appropriate amount of prototype knowledge. Both excessive and insufficient prototype knowledge negatively impact performance. Specifically, when the prototype knowledge is too few, the model tends to overly emphasize the commonalities between data, and neglect the distinctions within clusters. Conversely, an excessive amount of

prototype knowledge may cause the model to overlook the cluster-specific information, thereby failing to capture the commonalities within the same cluster. Based on our experimental results, we set $M$ to 7 for HandWritten dataset, while for other datasets, we set $M$ to 1. For more experimental results, please refer to Section C of the appendix.

### 4.7. Ablation Study

Our DSVC consists of three components: the knowledge aggregation learning module, the distribution consistency learning, and the knowledge guidance learning module. To assess the effectiveness of each component, we performed ablation studies on four versions of DSVC across eight different datasets. As shown in Tab.4, the removal of any component results in suboptimal performance. DSVC achieves optimal performance only when all three loss functions are included. The results demonstrate that the $L_r$ loss preserves the representative features of the data, maintaining its integrity and fidelity. The $L_a$ loss aligns the prototype knowledge distribution of the current view with the historical knowledge distribution, effectively mitigating the impact of concept drift. The $L_g$ loss leverages prototype knowledge to guide the data distribution of the current view, thereby enhancing the clustering structure.

### 4.8. Parameter Sensitivity Analysis

DSVC incorporates two trade-off parameters, $\alpha$ and $\beta$, which respectively regulate the reconstruction loss and the knowledge guidance learning loss. To evaluate the effectiveness of the trade-off parameters, we varied their values

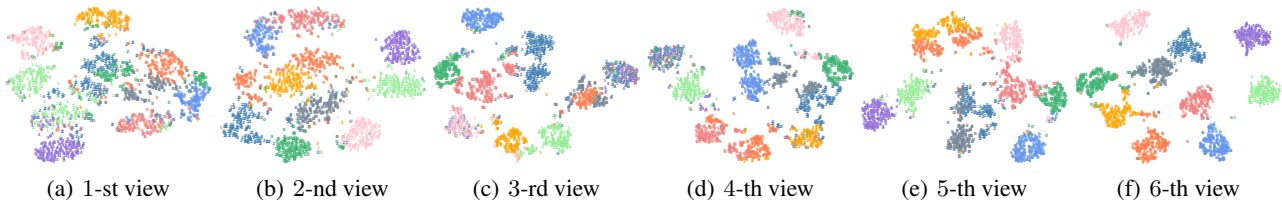

(a) 1-st view      (b) 2-nd view      (c) 3-rd view      (d) 4-th view      (e) 5-th view      (f) 6-th view

*Figure 7.* The visualization results of our DSVC on the HandWritten dataset as the view data accumulates.

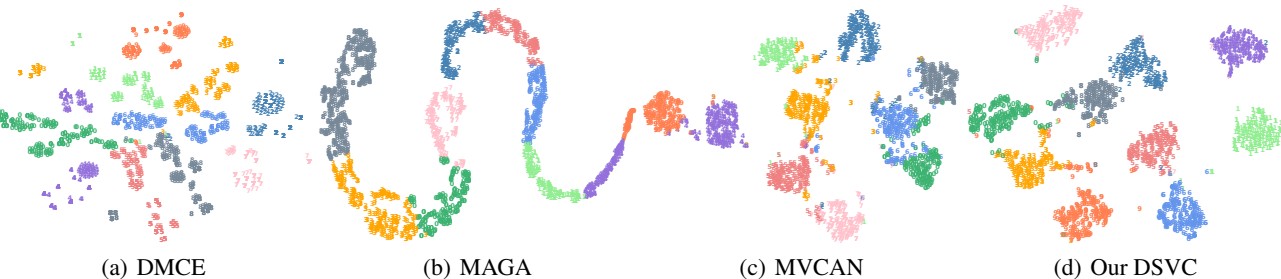

(a) DMCE      (b) MAGA      (c) MVCAN      (d) Our DSVC

*Figure 8.* The t-SNE visualization of representations learned by DMCE, MAGA, MVCAN, and our DSVC on the HandWritten dataset.

within the range of $10^2$ to $10^{-4}$. The experimental results, as illustrated in Fig.6, indicate that excessively high or low parameter values adversely impact clustering performance. This highlights the importance of balancing three losses within our model. Based on our experimental results, we recommend setting the parameter range between $10^1$ to $10^{-1}$. For more parameter analysis, see Section D of the appendix.

### 4.9. Visualization

To provide a more intuitive demonstration of the effectiveness of DSVC, we conduct visualization experiments on the Handwritten dataset. As shown in Fig.7, in the dynamic task, with the accumulation of views, the boundaries between clusters become increasingly distinct, and the intra-cluster structures become more compact. As illustrated in Fig.8, even when compared to DMVC methods within a static framework, our DSVC exhibits superior inter-cluster separability and intra-cluster compactness. This demonstrates that DSVC not only tackles the problem of concept drift between different view streams, but also enhances the clustering structure by leveraging the prototype knowledge of the current view to guide the distribution of view data.

## 5. Conclusion

In this paper, we propose a deep streaming view clustering (DSVC) method designed to address the challenges of concept drift and newly incoming view clustering in dynamic environments. To address these challenges, we employ the knowledge aggregation learning module to extract represen-

tative features and prototype knowledge from the current view data. Subsequently, to mitigate the effect of concept drift, we leverage the distribution consistency learning module to align the prototype knowledge of the current view with the historical knowledge distribution, to enhance the consistency of the distribution across different view streams. Furthermore, the knowledge guidance learning module is introduced to utilize prototype knowledge to guide data distribution, and strengthen the clustering structure. Finally, the prototype knowledge from the current view is updated into the knowledge base, which guides the training of subsequent views. Extensive experiments demonstrate the superiority and effectiveness of DSVC.

## Acknowledgments

This work was supported by the Sichuan Science and Technology Program (Grant nos. 2025ZNSFSC0474, 2024ZDZX0004 ), the Mianyang Science and Technology Program (Grant nos. 2023ZYDF091, 2023ZYDF003 ), the Sichuan Science and Technology Miaozi Program (Grant nos. MZGC20240057, MZGC20240144), and the Postgraduate Innovation Fund Project by Southwest University of Science and Technology (Grant no. 24ycx1010).

## Impact Statement

This paper presents work whose goal is to advance the field of Machine Learning. There are many potential societal consequences of our work, none of which we feel must be specifically highlighted here.

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

## A. DSVC Training Algorithm

In this section, we present the main algorithmic process of DSVC, as shown in Algorithm 1.

---

**Algorithm 1** The algorithm of DSVC

---

1: **Input:** New streaming view data $\mathbf{X}^v$ with size $N$; Batch size $S$; Training epoch $E$; The number of prototype knowledge is $K$.
2: **Parameter:** $\alpha$ and $\beta$.
3: **if** $v = 1$ **then**
4:     **while** $epoch < E$ **do**
5:         **for** $i = 1$ to $N/S$ **do**
6:             Training the network by Eq.(3).
7:             Aggregate prototype knowledge $\mathbf{P}^v$ and features $\mathbf{H}^v$ via Eq. (5).
8:             Optimize feature $\mathbf{H}^v$ distribution via Eq.(8).
9:             Leverages prototype knowledge $\mathbf{P}^v$ to guide the feature $\mathbf{H}^v$ distribution via Eq.(11).
10:         **end for**
11:         Update prototype knowledge $\mathbf{P}^v$ into knowledge base $\mathbf{B}$.
12:     **end while**
13:     Perform K-means algorithm on $\mathbf{H}^v$.
14: **end if**
15: **if** $v > 1$ **then**
16:     **while** $epoch < E$ **do**
17:         **for** $i = 1$ to $N/S$ **do**
18:             Training the network by Eq.(3).
19:             Aggregate prototype knowledge $\mathbf{P}^v$ and features $\mathbf{H}^v$ via Eq. (5).
20:             Align the current view prototype knowledge $\mathbf{P}^v$ with the knowledge base $\mathbf{B}$ distribution via Eq. (8).
21:             Leverages prototype knowledge $\mathbf{P}^v$ to guide the feature $\mathbf{H}^v$ distribution via Eq.(11).
22:         **end for**
23:         Update prototype knowledge $\mathbf{P}^v$ into knowledge base $\mathbf{B}$.
24:     **end while**
25:     Concatenate $\mathbf{H}^v$ to $\mathbf{H}^{v-1}$
26:     Perform K-means algorithm.
27: **end if**

---

## B. View Stream Analysis

In this section, we present additional experimental results regarding the continuous accumulation of views and the impact of different view streaming orders on performance, as shown in Figs.9 and 10. We can see from the Fig.9, that LSVC not only fails to address the concept drift problem but also cannot handle large-scale data. In contrast, our DSVC effectively mitigates concept drift while supporting the processing of large-scale data. This is achieved through our distribution consistency learning module, which aligns the knowledge distribution of the current view with historical knowledge, thereby alleviating the impact of concept drift. Additionally, as shown in Fig.10, the initial performance gap is quite significant when processing view streams of different sequences. However, as the views accumulate, the performance gap between different sequences gradually decreases. This is because the initially collected views exhibit significant differences in data distribution, which leads to a noticeable performance gap. As more views are accumulated, our distribution consistency learning module progressively aligns the data distribution, while the knowledge guidance learning module continuously enhances the clustering structure. As a result, the final performance tends to be stable and consistent.

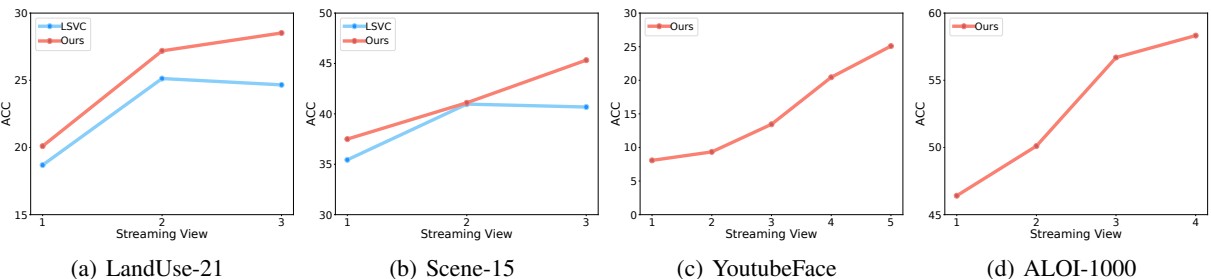

Figure 9. The changes of clustering performance on four datasets, when view data continue to accumulate.

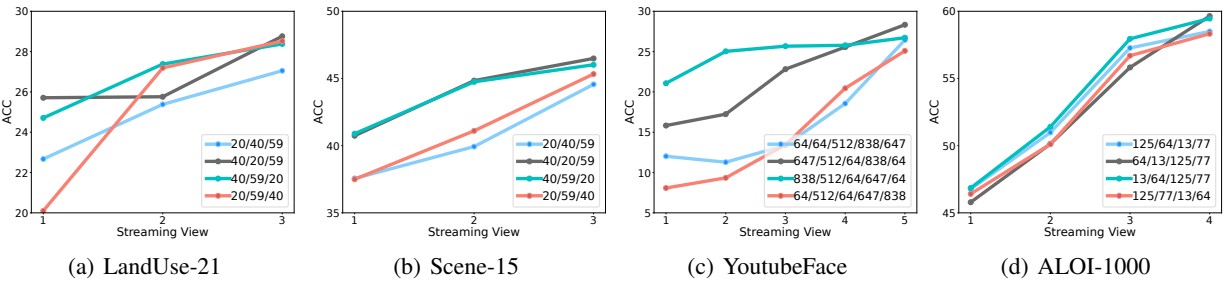

Figure 10. The clustering performance with different sequences of views streaming on four datasets

## C. Prototype Knowledge Number Analysis

Here are more experimental results on the effect of different numbers of prototype knowledge on the performance of the model. we conduct the experiment with prototype knowledge numbers ranging from $K$ to $M * K$, where $M = 1, 2, ..., 10$. As shown in Fig.11, the clustering performance can be enhanced with an appropriate amount of prototype knowledge. Both excessive and insufficient prototype knowledge negatively impact performance. Specifically, when the prototype knowledge is too few, the model tends to overly emphasize the commonalities between data, and neglect the distinctions within clusters. Conversely, an excessive amount of prototype knowledge may cause the model to overlook the cluster-specific information, thereby failing to capture the commonalities within the same cluster.

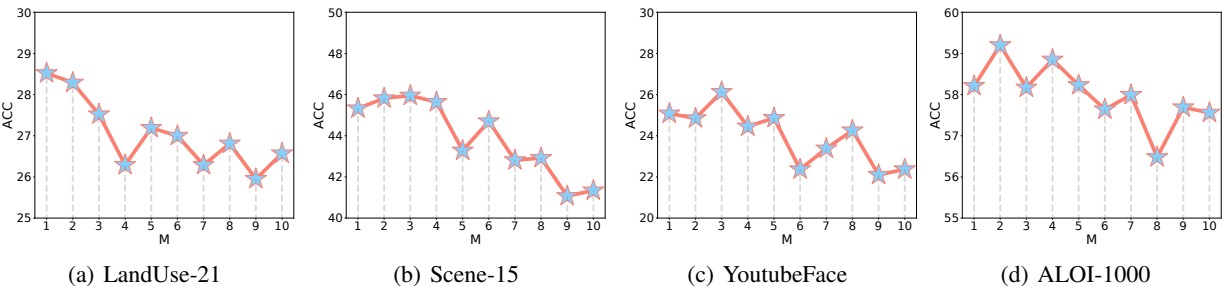

Figure 11. The clustering performance with different prototype knowledge numbers on four datasets

## D. Parameter Sensitivity Analysis

Here are more experimental results on the effect of different parameters. DSVC incorporates two trade-off parameters, $\alpha$ and $\beta$, which respectively regulate the reconstruction loss and the knowledge guidance learning loss. To evaluate the effectiveness of the trade-off parameters, we varied their values within the range of $10^2$ to $10^{-4}$. The experimental results, as illustrated in Fig.12, indicate that excessively high or low parameter values adversely impact clustering performance. This highlights the importance of balancing three losses within our model. Based on our experimental results, we recommend setting the parameter range between $10^1$ to $10^{-1}$.

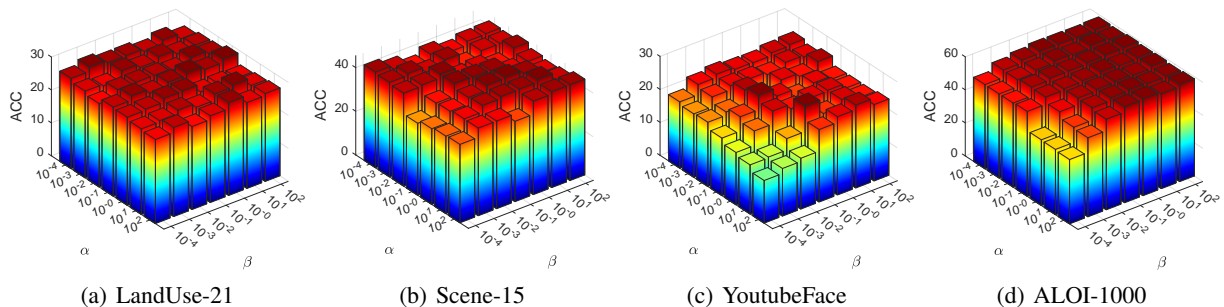

(a) LandUse-21       (b) Scene-15       (c) YoutubeFace       (d) ALOI-1000

*Figure 12.* Parameter sensitivity analysis on eight datasets.

# E. Limitations

This paper proposes a deep streaming view clustering (DSVC) method. To the best of our knowledge, DSVC is the first work to address stream view clustering within the deep learning framework, demonstrating superior performance even compared to DMVC methods under static frameworks. However, we only consider the scenario where concept drift exists between view streams, without considering the potential issue of noisy data that may occur in real-world data collection. In the future, we will focus on addressing and improving this issue.

