# OpenReview forum: "Deep Streaming View Clustering"
_ICML.cc/2025/Conference — ICML 2025 poster_

### Official Review · Reviewer_R49s · 2025-03-05

**Overall Recommendation:** 4

**Summary:**

This paper proposes a deep streaming view clustering algorithm (DSVC). It considers the scenario where data is acquired in the form of view streams in clustering tasks. DSVC aligns the prototype knowledge of the current view with the historical knowledge distribution, thereby mitigating the concept drift issue between streaming views. Furthermore, the aligned prototype knowledge guides the current data distribution, which enhances the clustering structure. Experimental results demonstrate that DSVC outperforms state-of-the-art methods.

**Claims And Evidence:**

Yes

**Essential References Not Discussed:**

[1] is also a recently proposed work to explore the clustering task when data views are collected overtime. This could be introduced and discussed.


[1] Wan et al. Fast Continual Multi-View Clustering With Incomplete Views. TNNLS 2024.

**Experimental Designs Or Analyses:**

Yes. I have checked the comparison to existing methods, ablation study and visualization analysis. They well support the claims.

**Methods And Evaluation Criteria:**

Yes. The metrics,  including ACC, NMI and ARI, are commonly used in clustering analysis.

**Other Comments Or Suggestions:**

Please see Section **Other Strengths And Weaknesses**

**Other Strengths And Weaknesses:**

Strengths:

1. This paper identifies and mitigates the issue of concept drift in streaming view clustering, which is novel in literature.

2. The proposed method achieves completing results in experiments.

3. The paper is well organized and easy to follow.

Weaknesses:

1. The authors mention the use of a cross-attention mechanism to reconstruct prototypes and features in Section 3.3, but there is no ablation study provided in the experiments to assess the impact of this component.

2. In Section 3.5 of the manuscript, the description of how the Knowledge Guidance Learning (KGL) module enhances the clustering structure need to be clarified more clearly.

3. As shown in Fig. 4, the training results of the first view exhibit significant differences depending on the view streams of the different sequences. Is there any method to reduce the discrepancies in the training results of the initial view?

**Questions For Authors:**

1.	What advantages does the proposed distribution consistency learning have over conventional contrastive learning strategies?
2.	Based on Tables 1 and 2, I noticed a relatively large standard deviation for the Stl10-fea dataset. Is this phenomenon expected or normal?

**Relation To Broader Scientific Literature:**

Existing multi-view methods assume all data views are collected in advance. However, in practical scenarios, new data views are collected over time. This paper proposes a DSVC method to solve this. This is meaningful and would have a broad impact in literature.

**Theoretical Claims:**

N/A. There is no theoretical claim.

---

> ### Author Rebuttal · Authors · 2025-04-01
>
> Thank you for taking the time to review our submission and provide valuable feedback.
>
> **W1: Ablation study without the cross-attention mechanism**
>
> **AW1:** To validate the effectiveness of the cross-attention mechanism, we conducted an ablation study. As shown in the following table, incorporating the cross-attention mechanism in our work can significantly enhance performance. This is because the cross-attention mechanism enables the prototype knowledge to better fit the data distribution while enhancing the intra-cluster consistency and inter-cluster discriminability of the features. The resulting prototype knowledge and features work in concert with our approach to achieve optimal performance.
>
> ### Ablation experiments on eight datasets (with or without cross-attention mechanism)
>
> | Datasets             | ALOI-10 | HandWritten | Landuse-21 | Scene-15 | stl10-fea | ALOI-100 | YoutubeFace-sel-fea | ALOI-1000 |
> |----------------------|---------|-------------|------------|----------|-----------|----------|---------------------|-----------|
> | **Cross-attention mechanism**  | acc     | acc         | acc        | acc      | acc       | acc      | acc                 | acc       |
> | ✗                    | 74.22   | 86.25       | 20.33      | 31.13    | 47.64     | 60.92    | 19.42               | 36.20     |
> | ✓                    | **86.69** | **95.44**  | **28.52**  | **45.33**| **63.41** | **80.53**| **25.09**           | **58.32** |
>
> ---
>
> **W2: In Section 3.5 of the manuscript, the description of how the Knowledge Guidance Learning (KGL) module enhances the clustering structure need to be clarified more clearly.**
>
> **AW2:** Our Knowledge Guidance module encourages samples to converge toward their corresponding prototypes while pushing away unrelated samples. This ensures that samples within the same cluster exhibit higher similarity, while those from different clusters exhibit lower similarity(i.e., intra-cluster compactness and inter-cluster separability.).
>
>
> **W3: Is there any method to reduce the discrepancies in the training results of the initial view?**
>
> **AW3:** Your question is highly insightful. In the experiments shown in Fig. 4, the first-view data in different streaming sequences originates from different sources, resulting in significant variations in data quality. Consequently, the training outcomes for the first view exhibit corresponding discrepancies. How to effectively exploit the information from the first-view data and enhance clustering performance in the absence of guidance information remains an open challenge. We will further investigate this issue in our future work.
>
> **Q1: What advantages does the proposed distribution consistency learning have over conventional contrastive learning strategies?**
>
> **AQ1:**
> Difference: Traditional contrastive learning aims to learn discriminative feature representations by pulling positive sample pairs closer while pushing negative pairs apart. Its primary objective is to enhance feature discriminability. In contrast, our Distribution Consistency Learning (DCL) measures the divergence between the probability distributions of two samples, with the goal of directly minimizing the discrepancy between distributions.
> Advantage: Traditional contrastive learning relies heavily on the construction of positive and negative sample pairs, where the quality and quantity of these pairs directly impact model performance. For instance, when there are noisy or low-quality samples (i.e., hard samples), the model can overly focus on these hard samples, which leads to overfitting and a lack of robustness. In contrast, our Distribution Consistency Learning (DCL) only requires measuring the similarity between sample distributions, eliminating the need for explicit positive and negative pair construction. This avoids performance degradation caused by low-quality negative samples, thereby enhancing the robustness of the model.
>
> **Q2: The standard deviation of Stl10-fea data set is relatively large. Is this phenomenon normal?**
>
> **AQ2:** As shown in Tables 1 and 2, the clustering performance on the Stl10-fea dataset exhibits a relatively large standard deviation across almost all methods. This is primarily due to the significant feature quality disparity between different views (it can be seen in Fig. 4), along with the large-scale sample but limited number of views. As a result, it becomes challenging to fully exploit the complementary information among views during training, leading to considerable performance fluctuations across different random initializations for all methods.

---

> > ### Comment · Reviewer_R49s · 2025-04-03
> >
> > Thanks for your responses. They have addressed my concerns.

---

### Official Review · Reviewer_WHyf · 2025-03-11

**Overall Recommendation:** 4

**Summary:**

This paper identifies the concept drift problem in streaming view clustering, which causes outdated models to fail to adapt to new view data. To address this, the authors employ knowledge aggregation learning to simultaneously reconstruct prototype knowledge from the features and reconstruct features from the prototype knowledge. To mitigate the concept drift between view streams, distribution consistency learning aligns the prototype knowledge of the current view with the historical knowledge, ensuring consistency in the data distributions across the collected views. Finally, knowledge guidance learning is introduced to leverage prototype knowledge to guide data distribution, thereby enhancing the clustering structure of feature representations.

**Claims And Evidence:**

The superior performance of DSVC and its effectiveness in mitigating concept drift are well-supported by extensive experimental evidence, including results across multiple datasets. Improvements in both accuracy and efficiency are clearly documented. However, the theoretical analysis and description of the DSVC mechanism, particularly the explanation of knowledge guidance learning, could be made clearer.

**Essential References Not Discussed:**

The paper cites key references; however, it would benefit from a more in-depth discussion on the connection with related techniques for handling concept drift.

**Experimental Designs Or Analyses:**

The experimental design is well-structured, with clear benchmarks and comparisons to other state-of-the-art (SOTA) methods. The paper analyzes the impact of different view stream sequences to validate the effectiveness of the proposed method in mitigating concept drift.

**Methods And Evaluation Criteria:**

The proposed deep streaming view clustering method is well-suited to address the streaming view clustering problem in multi-view clustering. The benchmark datasets and performance metrics used are appropriate for evaluating the effectiveness of the method. Through experimental design, DSVC is compared with existing methods, validating its effectiveness.

**Other Comments Or Suggestions:**

The methodology explanation and experimental analysis in this paper require a more detailed and comprehensive discussion from the authors.

**Other Strengths And Weaknesses:**

Strengths:

1. Due to knowledge aggregation learning, the obtained feature representations and prototype knowledge are more representative.
2. This paper employs distribution consistency learning to align the prototype knowledge of the current view with the historical knowledge distribution, effectively mitigating the concept drift issue.
3. The performance significantly outperforms the comparison methods.
4. The experiments are diverse and account for various types of data.

Weakness:

1. In this paper, the Knowledge Aggregation Learning module uses cross-attention mechanisms to reconstruct prototypes and features. This approach appears to have been used in other works as well. Could the authors clarify the novelty of its application in this context?
2. In Section 3.4, this paper proposes using the distribution consistency learning loss to ensure that the reconstructed features do not deviate from the original data distribution when processing the first view. However, no related experiments are presented to validate the effectiveness of this strategy. It is recommended to provide relevant experimental results to support this approach.
3. In Section 3.5, the explanation of why the clustering structure can be enhanced is not very clear.
4. In the last paragraph of Section 4.4, the explanation for why the method can effectively handle large-scale datasets is not comprehensive.
5. In this paper, the training of subsequent views appears to depend on the previously trained views. Would the quality of the first view negatively impact the training of subsequent views if the first view is of poor quality?
6. In Fig. 3 and 4, it can be observed that in certain datasets, there is a sudden performance improvement after training on a particular view. Please explain the relevant reasons.
7. In Table 2 and Fig. 11, it is observed that the performance on the Stl10-fea dataset exhibits significant fluctuations. Is this behavior due to the inherent characteristics of the dataset, or is there another underlying reason?

**Questions For Authors:**

For related issues in this section, please refer to the Weakness.

**Relation To Broader Scientific Literature:**

This paper positions itself within the context of streaming view clustering, comparing DSVC with other multi-view clustering methods. It highlights the lack of the research regarding streaming view clustering and proposes a novel approach for addressing the streaming view clustering.

**Theoretical Claims:**

The theoretical claims regarding streaming view clustering and the necessity of addressing concept drift between view streams are well-supported. However, the mathematical formulation of DSVC could benefit from greater clarity. Specifically, the definition of how knowledge guidance learning enhances the clustering structure is lacking in terms of formal mathematical work.

---

> ### Author Rebuttal · Authors · 2025-04-01
>
> **W1: The cross-attention mechanism appears to have been used in other works as well. Could the authors clarify the novelty of its application in this context?**
>
> **AW1:** Other methods, such as ProImp [1], are designed for static multi-view clustering tasks, where the attention mechanism primarily focuses on one-time data reconstruction and clustering. In contrast, DSVC is designed for streaming multi-view clustering scenarios, where view data arrives dynamically over time and is subject to concept drift. The Knowledge Aggregation Learning (KAL) module continuously processes newly arrived views and interacts dynamically with the historical knowledge repository to maintain global consistency in data distribution. The KAL module in this dynamic environment fundamentally differs from conventional objectives.
>
> **Reference:**
>
> [1] Incomplete multi-view clustering via prototype-based imputation, IJCAI, 2023.
>
> **W2:The distributed consistency loss used in the first view of this paper lacks a correlation ablation experiment.**
>
> **AW2:** We have added ablation experiments for the Distribution Consistency Loss (DCL). As shown in the table, incorporating DCL in the first view significantly enhances clustering performance.
>
> | Datasets       | ALOI-10 | HandWritten | Landuse-21 | Scene-15 | stl10-fea | ALOI-100 | YoutubeFace-sel-fea | ALOI-1000 |
> |----------------|---------|-------------|------------|----------|-----------|----------|---------------------|-----------|
> | **DCL loss**   | acc     | acc         | acc        | acc      | acc       | acc      | acc                 | acc       |
> | ✗              | 79.89   | 88.05       | 26.52      | 42.91    | 44.71     | 78.55    | 21.50               | 50.13     |
> | ✓              | **86.69** | **95.44**  | **28.52**  | **45.33**| **63.41** | **80.53**| **25.09**           | **58.32** |
>
> ---
>
> **W3: In Section 3.5, the explanation of why the clustering structure can be enhanced is not very clear.**
>
> **AW3:** The knowledge guidance learning loss computes the similarity relationships between prototypes and samples, firstly.  Then, it maximizes the similarity between features and their corresponding prototypes while minimizing the similarity between features and prototypes of different classes, thereby reinforcing the intra-cluster feature consistency and the inter-cluster feature discriminability.
>
> **W4: The explanation for why DSVC can effectively handle large-scale datasets is not comprehensive.**
>
> **AW4:** Our DSVC leverages a prototype knowledge base to transfer historical knowledge, which considers only the number of prototypes $K$ and current-view samples $n$ during computation. Thus, the computational complexity of $O(K \times n)$, where $(K \ll n)$.   Moreover, the memory overhead during updates is significantly reduced to $O(K \times d)$, where $d$ denotes the embedding dimension. Therefore, our method can effectively process large-scale data under limited conditions.
>
> **W5: Would the quality of the first view negatively impact the training of subsequent views?**
>
> **AW5:** As shown in Fig. 4 and 9, the quality of the first view does not have a significant negative impact on subsequent training. For example, in the HandWritten dataset, despite substantial variations in the quality of the first view, the final results remain relatively stable as more view data is continuously collected. Even when the initial view quality is suboptimal, the overall performance difference remains minimal.
>
> **W6: In Fig. 3 and 4, it can be observed that there is a sudden performance improvement after training on a particular view.**
>
> **AW6:** This phenomenon occurs because different views are collected from distinct sources, resulting in significant variations in feature quality across views. Consequently, during training, when a view with highly distinguishable feature representations (i.e., a view with high inter-class separability) is encountered, the performance experiences a noticeable boost.
>
> **W7: In Table 2 and Fig. 11, it is observed that the performance on the Stl10-fea dataset exhibits significant fluctuations.**
>
> **AW7:**
>
> **Large standard deviation:** This is primarily due to the significant feature quality disparity between different views (feature dimensions are 512, 1024, and 2048, respectively) of the Stl10-fea dataset, along with the large-scale sample but limited number of views. As a result, it becomes challenging to fully exploit the complementary information among views during training, which leads to considerable performance fluctuations for all methods.
>
> **Performance fluctuations:** Our method is sensitive to the parameters $\alpha $ and $\beta $. To intuitively investigate the impact of $ \alpha $ and $ \beta $ on performance, we conducted the sensitivity analysis presented in Fig. 11. In our approach, we set $ \alpha $ and $ \beta $ to 0.001 and 1, respectively, for the STL10-fea dataset.

---

> > ### Comment · Reviewer_WHyf · 2025-04-03
> >
> > Thank you for the detailed response, which overcomes my concerns. I maintain my score.

---

### Official Review · Reviewer_Rmer · 2025-03-11

**Overall Recommendation:** 3

**Summary:**

The paper presents Deep Streaming View Clustering (DSVC), a method designed to tackle concept drift in multi-view clustering with streaming data. DSVC features three modules: Knowledge Aggregation Learning (KAL) for feature extraction, Distribution Consistency Learning (DCL) to align current and historical knowledge, and Knowledge Guidance Learning (KGL) to enhance clustering. It outperforms 12 state-of-the-art methods in clustering accuracy, stability, and scalability on various datasets, making it effective for real-world streaming data applications.

**Claims And Evidence:**

The authors claims there exists the data imbalance of distribution between different view streams, i.e. concept drift problem. Figure 1(a) needs more detailed explanation about the model and evaluation settings. Also, it’s better to explain how this concept drift problem affect the performance in theory rather than just the experimental results.

**Essential References Not Discussed:**

No.

**Experimental Designs Or Analyses:**

Yes, the experimental designs and analyses are valid.

**Methods And Evaluation Criteria:**

Yes.

**Other Comments Or Suggestions:**

No.

**Other Strengths And Weaknesses:**

Strengths
- The paper is well-constructed and easy to read.
- The experiments are conducted over 8 datasets and 12 state-of-the-art methods.

Weaknesses
- The authors should include the notation of Z and H in Figure 2 for better understanding.
- The authors claim they might be the first work to reveal the issue of concept drift in the context of streaming view clustering, but the paper lacks thorough investigation to address the problem.
- In section 3.1, P is of the dimension K*d, but in section 3.4, P seems to have the same dimension as B, N*K. Also, it would be better to clarify the dimensions of the later-introduced variable, such as p, h, b in the context for better understanding.

**Questions For Authors:**

please refer to the weakness part.

**Relation To Broader Scientific Literature:**

In summary, the paper makes an important contribution by bridging the gap between static multi-view clustering methods and dynamic streaming clustering methods. Its focus on concept drift, distribution consistency, and the alignment of historical and current knowledge opens up new possibilities for more effective clustering in continuously evolving datasets. These contributions are well-aligned with the broader scientific efforts to improve streaming learning and multi-view clustering.

**Theoretical Claims:**

Eq. 7 needs further explanation about the historical prototype self-similarity term in the denominator. It appears that the probability $Q_i^b$ cannot sum up to 1 if this term exists, and the authors do not explain why this term exists.

---

> ### Author Rebuttal · Authors · 2025-04-01
>
> We sincerely appreciate the time you have taken to review our submission and provide valuable feedback.
>
> **Q1: Claims And Evidence**
>
> **Q1(a): Figure 1(a) needs more detailed explanation about the model and evaluation settings.**
>
> **AQ1(a):** Figure 1 (a) illustrates that due to distribution imbalance across different views, i.e., the concept drift problem, the performance of different views also varies.
>
> **Q1(b): How does this concept drift problem affect the performance?**
>
> **AQ1(b):** Thank you for your thoughtful comments! Since multi-view data are collected from different sensors or captured from various views, discrepancies in distribution and quality naturally arise among different views. This phenomenon is referred to as the concept drift problem. Existing multi-view clustering methods assume that all views are pre-collected for joint training simultaneously. However, in dynamic environments, due to the mechanisms of data generation, sampling frequencies and transmission speeds vary across different views. As a result, multi-view data is typically collected in a sequential manner (i.e., streaming view), which makes it infeasible to wait until all the views are ready for training. Therefore, data must be processed sequentially in a streaming fashion, training one view at a time. However, due to the presence of concept drift across different view streams, the similarity relationships between different samples and the feature representations of the same sample may vary across views after model training, which is shown in the following image (The image is in the anonymous URL: https://anonymous.4open.science/r/Fig-0D5B). This discrepancy reduces the intra-class feature consistency and the inter-class feature discriminability, thereby degrading clustering performance.
>
> **Q2: Theoretical Claims: The probability $Q_i^b$ cannot sum up to 1 if Eq.7 exists, and the authors do not explain why this term exists.**
>
> **AQ2:** Thank you very much for your insightful comment! The derived $Q_i^b$ from Eq. 7 does not sum to 1. Therefore, we apply the normalization to ensure its sum is strictly 1, making it suitable for the computation of $\mathcal{L}_{d}$. Eq. 7 computes the distribution probability of class-specific prototypes in both historical knowledge and the current view. The complete formulation is as follows:
>
> $$
> Q_i^b =\frac{\exp{(S(p_i^v,b_i)/\tau)}}{\sum_{j=1}^{K}\exp{(S(p_i^v,b_i)/\tau)}+{\sum_{j\neq i}^{K}\exp{(S(b_i,b_j)/\tau)}}},Q_i^p =\frac{\exp{(S(b_i,p_i^v)/\tau)}}{\sum_{j=1}^{K}\exp{(S(b_i,p_i^v)/\tau)}+{\sum_{j\neq i}^{K}\exp{(S(p_i,p_j)/\tau)}}}.
> $$
>
> Therefore, $ Q_i^b $ can be interpreted as the similarity probability of the $ i $-th prototype knowledge ($ b_i $ and $ p_i^v $) mapped within the historical knowledge base ($ B $). Similarly, $ Q_i^p $ represents the similarity probability of $ b_i $ and $ p_i^v $ mapped within the current view knowledge. We achieve alignment of the current view prototype knowledge with the historical knowledge distribution by minimizing the discrepancy between the $ Q_i^b $ and $ Q_i^p $ distributions.
>
> **W1: The authors should include the notation of Z and H in Figure 2 for better understanding.**
>
> **AW1:** We have optimized Figure 2 accordingly in the revised manuscript.
>
> **W2: The authors claim they might be the first work to reveal the issue of concept drift in the context of streaming view clustering, but the paper lacks thorough investigation to address the problem.**
>
> **AW2:** Through a comprehensive survey and review of related literature, the most comparable methods to our setup are CAC [1], ACMVC [2], LSVC [3], and OBAL [4]. Among them, CAC and LSVC achieve streaming view clustering by maintaining a consensus matrix, while ACMVC and OBAL consider data stream scenarios. However, none of these methods account for the concept drift problem in streaming views, highlighting the novelty of our proposed DSVC. Furthermore, we have conducted a more comprehensive literature analysis to enrich the manuscript. We look forward to further discussions with you.
>
> **Reference：**
>
> [1] Live and learn: Continual action clustering with incremental views, AAAI, 2024.
>
> [2] Continual multi-view clustering with consistent anchor guidance, IJCAI 2024
>
> [3] LSVC: A Lifelong Learning Approach for Stream-View Clustering, TNNLS, 2024.
>
> [4] Online boosting adaptive learning under concept drift for multistream classification, AAAI, 2024.
>
> **W3: The dimensions of P and B need to be clarified.**
>
> **AW3:** We have revised the ambiguous parts of the manuscript to enhance clarity. Specifically, we have updated $\mathbf{B}=\{b_1,b_2,\ldots,b_K\}\in{ \mathbb {R}}^{N \times K}$ to $\mathbf{B}=\{b_1,b_2,\ldots,b_K\}\in{ \mathbb {R}}^{K \times d}$ for better comprehension.

---

> > ### Comment · Reviewer_Rmer · 2025-04-09
> >
> > Thank you for your detailed response. I have also read the comments from other reviewers. Most of my concerns have been adequately addressed.  As a result, I would like to keep my score as '3', leading to acceptance.

---

### Official Review · Reviewer_oVPf · 2025-03-12

**Overall Recommendation:** 4

**Summary:**

This paper explores a rarely addressed area in multi-view clustering, namely, streaming view clustering. In this work, the authors utilize the Knowledge Aggregation Learning (KAL) module to extract features and prototype knowledge. Subsequently, the Distribution Consistency Learning (DCL) module is employed to mitigate the concept drift problem across view streams. Finally, the Knowledge Guidance Learning (KGL) module is introduced to enhance the clustering structure. Extensive experiments demonstrate the effectiveness of the proposed method.

**Claims And Evidence:**

Yes, the superiority and effectiveness of the proposed method are supported by extensive experimental evidence.

**Essential References Not Discussed:**

In this paper, each theory and technique is supported by relevant references.

**Experimental Designs Or Analyses:**

Yes, the paper conducts extensive experiments to demonstrate the effectiveness of the proposed method. By comparing it with state-of-the-art DMVC and SVC methods, the superiority of its performance is highlighted. The effectiveness of the method in mitigating concept drift is further validated through experiments with varying view stream sequences. Additional ablation studies also substantiate the rationality and effectiveness of each component.

**Methods And Evaluation Criteria:**

Yes, the proposed method effectively addresses the concept drift issue between view streams. The benchmark datasets (e.g., ALOI-10, HandWritten, LandUse-21, etc.) and performance metrics (ACC, NMI, ARI) used are appropriate for evaluating the effectiveness of the method.

**Other Comments Or Suggestions:**

1) The authors should clarify the meaning of each symbol used in the manuscript. For instance, in Eq. 4, what does the symbol $d$ represent? Is it distance or dimensionality?
2) The authors are advised to ensure consistent use of capitalization throughout the manuscript. For example, in the first paragraph of Section F in the appendix, $Loss$ is capitalized, whereas it is written in lowercase in other sections.

**Other Strengths And Weaknesses:**

This paper demonstrates strong originality by identifying the issue of concept drift in the context of view streams. The proposed method is novel, with the distribution consistency learning module effectively mitigating concept drift between view streams. Comprehensive experiments are conducted to validate the superiority and effectiveness of the proposed approach. However, some deeper explanations are lacking in both the theoretical interpretation and experimental analysis. Further in-depth theoretical and experimental exploration would benefit the paper.

**Questions For Authors:**

Q1: As far as I know, domain streams and view streams share conceptual similarities, and the authors have mentioned domain stream learning in related work. Could you clarify the distinction between domain streams and view streams?
Q2: In Section 3.3, the authors employ a cross-attention mechanism to learn prototypes. What are the advantages of this approach compared to using prototypes learned through k-means clustering?
Q3: In the last paragraph of Section 4.4, the authors mention that some DMVC methods cannot handle large-scale data under limited computational resources.  However, we observe that the stream-view clustering method LSVC also struggles with large-scale data, but the authors do not provide an explanation for this phenomenon.
Q4: During clustering, are the features used those derived from attention-based reconstruction, or the features extracted by the autoencoder?
Q5: Based on Figure 11, we observe that in some datasets (such as Stl10-fea and YoutubeFace-sel-fea), the parameters $\alpha$ and $\beta$ have a significant impact on the results. Could the authors provide an explanation for why this phenomenon occurs?

**Relation To Broader Scientific Literature:**

This study considers the scenario of data acquisition via view streams under dynamic conditions and compares the proposed method with existing DMVC and SVC approaches. It highlights the gap in the field of streaming view clustering and introduces a novel DSVC method to address these issues.

**Theoretical Claims:**

Yes, in this paper, the necessity of addressing the concept drift problem in stream-based view clustering is thoroughly supported by relevant literature and experimental evidence provided by the authors.

---

> ### Author Rebuttal · Authors · 2025-04-01
>
> Firstly, we sincerely thank you for your detailed review and constructive feedback, which have greatly contributed to improving the presentation of our submission.
>
> **S1: For instance, in Eq. 4, what does the symbol d represent?**
>
> **AS1:** Thank you very much for your valuable suggestions. In Eq. (4), $d$ represents the dimensionality of the feature $Z$.
>
> **S2:The authors are advised to ensure consistent use of capitalization throughout the manuscript.**
>
> **AS2:** We have already carefully reviewed the entire paper for consistency in terminology to ensure that the final version does not have such problems.
>
> **Q1: Could you clarify the distinction between domain streams and view streams?**
>
> **AQ1:** **Domain Stream** means that when a learner faces a series of tasks, the data input distribution (domain) for each task may be different, but the set of categories for the tasks remains the same. **View Stream** refers to a scenario where multiple views of the same object arrive sequentially in a streaming manner, one after another. The key distinction between the two lies in the difference of their task: **Domain Stream** primarily tackles domain drift, continual adaptation, and mitigating catastrophic forgetting while maintaining task consistency. In contrast, **View Stream** focuses on addressing challenges such as collaborative learning across multiple views and effectively handling view heterogeneity.
>
> **Q2: What are the advantages of the cross-attention mechanism compared to using prototypes learned through k-means clustering?**
>
> **AQ2:**
>
> **Advantage 1:** **$k$-means** partitions cluster centers based on a fixed distance metric. Its prototypes (i.e., cluster centers) are static and cannot dynamically adjust their distribution according to the data characteristics. In contrast, the prototypes learned through our **cross-attention mechanism** can dynamically adjust based on the data distribution, allowing the prototype knowledge to better fit with the data and become more representative.
>
> **Advantage 2:** The prototypes learned by **$k$-means** are susceptible to the influence of outliers (or noise). In contrast, the **cross-attention mechanism**, through the collaborative optimization between features and prototype knowledge, ensures that the learned prototypes are less affected by outliers (or noise), thereby enhancing robustness.
>
> **Q3: We observe that the stream-view clustering method LSVC also struggles with large-scale data, but the authors do not provide an explanation for this phenomenon.**
>
> **AQ3:** Thank you for your constructive feedback. Our DSVC leverages a prototype knowledge base to transfer historical knowledge, considering only prototypes and current-view samples during computation. This results in a computational complexity of $O(K \times n)$, where $K$ is the number of prototypes and $n$ is the number of samples ($(K \ll n$). Moreover, the memory overhead during updates is significantly reduced to $O(K \times d)$, where $d$ denotes the embedding dimension. In contrast, LSVC exhibits a computational complexity of $O(n^2)$ and maintains historical knowledge via a consensus bipartite graph $\mathbf{Z} \in \mathbb{R}^{k \times n}$, requiring a storage overhead of $O(k \times n)$. As the data scale increases, the computation and update of $\mathbf{Z}$ lead to a substantial surge in memory consumption. Moreover, LSVC is a traditional shallow method, whereas DSVC is a deep learning-based approach. As a result, DSVC is well-suited for handling large-scale datasets, while LSVC struggles to cope with such scenarios.
>
> **Q4: During clustering, are the features used those derived from attention-based reconstruction, or the features extracted by the autoencoder?**
>
> **AQ4:** We perform clustering using the features $ H $ reconstructed through the cross-attention mechanism.
>
> **Q5. Fig. 11, it observe that in some datasets, the parameters $\alpha$ and $\beta$ have a significant impact on the results.**
>
> **AQ5:** Our method exhibits a certain degree of sensitivity to the parameters $ \alpha $ and $ \beta $. To analyze this effect, we conducted a sensitivity analysis, as shown in Fig.11. The results indicate that excessively high or low parameter values adversely impact clustering performance. This highlights the importance of balancing three losses within our model. In our approach, for the STL10-fea dataset, we set $ \alpha $ and $ \beta $ to 0.001 and 1, respectively, and set to 0.1 for both $ \alpha $ and $ \beta $ for the ALOI-1000 and ALOI-100 datasets. For the remaining datasets, $ \alpha $ and $ \beta $ are set to 0.1 and 1, respectively.

---

### Decision · Program_Chairs · 2025-05-01

**Decision:**

Accept (poster)

**Comment:**

All Reviewers agree on the Acceptance of this paper.